# Adverse childhood experiences, adult depression, and suicidal ideation in rural Uganda: A cross-sectional, population-based study

Emily N. Satinsky[1,2]*, Bernard Kakuhikire[1], Charles Baguma[1], Justin D. Rasmussen[3], Scholastic Ashaba[1], Christine E. Cooper-Vince[4], Jessica M. Perkins[5], Allen Kiconco[1], Elizabeth B. Namara[1], David R. Bangsberg[1,6], Alexander C. Tsai[1,2,7]

1 Mbarara University of Science and Technology, Mbarara, Uganda, 2 Center for Global Health, Massachusetts General Hospital, Boston, Massachusetts, United States of America, 3 Duke University, Durham, North Carolina, United States of America, 4 Départment de Psychiatrie, Universitié de Genève, Switzerland, 5 Peabody College, Vanderbilt University, Nashville, Tennessee, United States of America, 6 Oregon Health and Science University—Portland State University School of Public Health, Portland, Oregon, United States of America, 7 Harvard Medical School, Boston, Massachusetts, United States of America

* esatinsky@mgh.harvard.edu

**Data Availability Statement:** All files are available from the following GitHub repository: https://github.com/esatinsky/acesdepression_paper.

## Abstract

### Background

Depression is recognized globally as a leading cause of disability. Early-life adverse childhood experiences (ACEs) have been shown to have robust associations with poor mental health during adulthood. These effects may be cumulative, whereby a greater number of ACEs are progressively associated with worse outcomes. This study aimed to estimate the associations between ACEs and adult depression and suicidal ideation in a cross-sectional, population-based study of adults in Uganda.

### Methods and findings

Between 2016 and 2018, research assistants visited the homes of 1,626 adult residents of Nyakabare Parish, a rural area in southwestern Uganda. ACEs were assessed using a modified version of the Adverse Childhood Experiences-International Questionnaire, and depression symptom severity and suicidal ideation were assessed using the Hopkins Symptom Checklist for Depression (HSCL-D). We applied a validated algorithm to determine major depressive disorder diagnoses. Overall, 1,458 participants (90%) had experienced at least one ACE, 159 participants (10%) met criteria for major depressive disorder, and 28 participants (1.7%) reported suicidal ideation. We fitted regression models to estimate the associations between cumulative number of ACEs and depression symptom severity (linear regression model) and major depressive disorder and suicidal ideation (Poisson regression models). In multivariable regression models adjusted for age, sex, primary school completion, marital status, self-reported HIV status, and household asset wealth, the cumulative

**Funding:** This project was funded by Friends of a Healthy Uganda and U.S. National Institutes of Health R01MH113494-01 awarded to ACT (https://projectreporter.nih.gov/project_info_description.cfm?aid=9507908&icde=43069576&ddparam=&ddvalue=&ddsub=&cr=1&csb=default&cs=ASC&pball=). The funders had no role in study design, data collection and analysis, decision to publish, or preparation of the manuscript.

**Competing interests:** I have read the journal's policy and the authors of this manuscript have the following competing interests: ACT receives a stipend as a Specialty Consulting Editor for PLoS Medicine and serves on the journal's editorial board.

**Abbreviations:** ACE, adverse childhood experience; ACE-IQ, Adverse Childhood Experiences–International Questionnaire; ARR, adjusted relative risk; CASIC, Computer-Assisted Survey Information Collection; CI, confidence interval; DSM, Diagnostic and Statistical Manual of Mental Disorders; HICs, high-income countries; HSCL-D, Hopkins Symptom Checklist for Depression; IQR, interquartile range; LMICs, low- and middle-income countries; SD, standard deviation; STROBE, Strengthening the Reporting of Observational Studies in Epidemiology.

number of ACEs was associated with greater depression symptom severity (b = 0.050; 95% confidence interval [CI], 0.039–0.061, $p < 0.001$) and increased risk for major depressive disorder (adjusted relative risk [ARR] = 1.190; 95% CI, 1.109–1.276; $p < 0.001$) and suicidal ideation (ARR = 1.146; 95% CI, 1.001–1.311; $p = 0.048$). We assessed the robustness of our findings by probing for nonlinearities and conducting analyses stratified by age. The limitations of the study include the reliance on retrospective self-report as well as the focus on ACEs that occurred within the household.

## Conclusions

In this whole-population, cross-sectional study of adults in rural Uganda, the cumulative number of ACEs had statistically significant associations with depression symptom severity, major depressive disorder, and suicidal ideation. These findings highlight the importance of developing and implementing policies and programs that safeguard children, promote mental health, and prevent trajectories toward psychosocial disability.

## Author summary

### Why was this study done?

- Depression is recognized globally as a leading cause of disability. Studies from high-income countries have shown robust associations between adverse childhood experiences (ACEs) and depression during adulthood.

- While studies from sub-Saharan Africa have demonstrated associations between ACEs and depression and suicidality among children, adolescents, and young adults, no study from this region has yet estimated the associations between ACEs and major depressive disorder and suicidal ideation within a whole-population sample of adults.

### What did the researchers do and find?

- We conducted a cross-sectional, population-based study of 1,626 adults in rural Uganda, eliciting ACEs, current depression, and suicidal ideation through face-to-face interviews.

- The cumulative number of ACEs that occurred before age 18 had statistically significant associations with adult depression symptom severity, major depressive disorder, and suicidal ideation.

- Depression symptom severity and major depressive disorder had statistically significant associations with each of the 9 types of ACEs. Suicidal ideation also had statistically significant associations with living with an adult who was sent to jail or prison during childhood and experiencing food and/or water insecurity during childhood.

**What do these findings mean?**

- Our interpretation of these findings raises implications for the development of policies and programs that support children, adolescents, and their families, and promote mental health.

- We are not able to determine the extent to which these associations are causal, and our analysis is susceptible to potential bias from the use of retrospective self-report of ACEs.

## Introduction

Major depressive disorder, which is characterized by a range of symptoms such as depressed mood, feelings of low self-worth, anhedonia, and decreased energy [1], has been recognized globally as a leading cause of disability [2]. Associated with high rates of morbidity and mortality, depression negatively affects individuals' social, occupational, and physical functioning [3]. In low- and middle-income countries (LMICs), depression is the leading neuropsychiatric cause of the burden of disease [4], and by 2030, depression is projected to be the leading cause of the global burden of disease [5]. In addition to its adverse effects on quality of life and functioning, depression has consistently been shown to be a strong risk factor for suicide [6].

Adverse childhood experiences (ACEs) include a range of early-life challenges and traumatic events that occur before age 18 and may put an individual at risk for negative outcomes throughout the life course [7]. Such experiences include emotional, physical, and sexual abuse; household dysfunction; and neglect. Research studies from high-income countries (HICs) have shown robust associations between ACEs and adverse mental health outcomes in adulthood, including antisocial behaviors [8], adult life stress [9], smoking [10], problematic substance use [11], depression [12], and suicide [13]. Research has found that ACEs increase children and adolescents' risk of having low resilience factors [14]. The magnitude of this association may partially depend on individual-level vulnerability. A study from Australia directly testing the diathesis-stress model for depression found that adults who had high predispositional vulnerability and who had experienced more stressful, adverse life experiences were at the highest risk of developing depression [15].

The effects of ACEs on adult health outcomes may be cumulative, whereby individuals who experience more ACEs have greater risk for developing mental ill health as adults [13,16]. The cumulative disadvantage theory posits that early advantage and disadvantage, including that which results from genetic and environmental factors, may compound, resulting in markedly differing trajectories over time [17]. In addition to the cumulative nature of ACEs, some experiences may be more salient than others; one study, for example, found stronger associations between negative mental health outcomes and child maltreatment than with household dysfunction [18]. An umbrella review of 19 meta-analyses found strong associations between childhood sexual abuse and depression in adulthood, as well as other psychosocial and psychiatric outcomes [19].

Childhood adversity is prevalent across LMICs, with evidence demonstrating the exacerbating effects of poverty and family violence on other childhood traumas [20]. Negative socioeconomic conditions in general, including insecure access to food and water, present additional obstacles to healthy development, with documented negative effects on school attendance, health, and well-being [21]. Prior research from other contexts also indicate that neighborhood

poverty is positively associated with abuse, child maltreatment [22,23], and household dysfunction including high caretaker stress and depression [24]. Living in unsafe, "stressogenic" environments characterized by economic precarity, a lack of resources, and violence increases children and adolescents' exposure to stress and trauma [25]. Furthermore, as "poverty begets trauma" [25] and "stress begets stress" [26], individuals who grow up in such environments are at a higher risk of experiencing psychopathology as adults [27,28]. Thus, the pervasive poverty in some LMICs further underscores the importance of studying the relationship between ACEs and adult mental ill health.

While research studies from sub-Saharan Africa indicate a high prevalence of childhood adversity [29–32], research on the associations between ACEs and adult depression and suicidal ideation in these settings is limited. A study from South Africa found that emotional neglect and sexual abuse before age 18 were both associated with suicidality, problematic substance use, and depression 2 years after initial measurement among adolescents and young adults aged 15 to 26 [30]. Depression assessment in this study was based on past-month symptom screening using the Centre for Epidemiologic Studies Depression Scale. Similarly, a prospective study among South African adolescents found a statistically significant, graded relationship between the cumulative number of ACEs and suicide behaviors after adjusting for baseline suicidality [33]. Two studies of children and adolescents in Uganda found that depression was associated with loss of a parent and alcohol consumption [34] and domestic violence [35]. These and other studies have indicated a high prevalence of ACEs and associations with depression and suicidality among children, adolescents, and young adults in sub-Saharan Africa [31,32,36,37]. To address the gap in the literature on ACEs and mental health among adults, we aimed to estimate the associations between ACEs and depression symptom severity, major depressive disorder, and suicidal ideation in a population-based sample of adults in rural Uganda.

## Methods

This study is reported following the Strengthening the Reporting of Observational Studies in Epidemiology (STROBE) guideline (S1 Checklist).

### Study setting and population

This cross-sectional study took place in Mbarara District, a rural region of southwestern Uganda, as part of a population-wide longitudinal study [38]. Mbarara District is made up of 16 subcounties, 90 parishes, and 910 villages. In 2014, the population was estimated at 474,144 people [39]. Through an iterative process involving field site visits and informal conversations with local leaders and other prominent village residents, our team selected Nyakabare Parish as the study site. It was smaller than other parishes in the region, which facilitated our team's ability to capture a whole-population sample; the local leaders were supportive of engaging area residents in a population-based household survey; and the local leaders stated that there was relatively less nongovernmental organization involvement in the area, in terms of service delivery or other development activities. Thus, the present study took place in the 8 villages of Nyakabare Parish, about 20 kilometers outside of Mbarara Town. Parish residents commonly make their livelihoods from subsistence farming and animal husbandry, and income is often supplemented by migratory work; food and water insecurity are common [40–42].

The national Ugandan Violence Against Children Survey estimated that 35% of girls experience sexual violence and 68% of boys experience physical abuse during childhood and adolescence [43]. Additionally, research studies from across Uganda have estimated a prevalence of probable depression among adults between 17% and 29% [44–46]. Despite the high prevalence

of depression in Uganda, there are severe resource shortages for mental health care [47]. Only 1% of the overall health budget in Uganda is allocated to mental health [48]. Furthermore, there is currently less than 1 mental health provider for every 100,000 people in the country [49], and there are limited treatment resources for substance use [50,51]. Resource limitations, coupled with poverty and stigma toward mental illness [52], contribute to a wide treatment gap [53].

## Sampling procedure and data collection

Prior to starting the study, a population census was conducted, including the enumeration of all 1,933 adults across the 758 households in the parish. All adults 18 years and older, and emancipated minors between 16 and 18 years, who reported stable residence in the parish were considered for study participation. Emancipated minors were defined as persons younger than 18 years who are married, are pregnant, live with a biological child in the household, or are responsible for their own livelihood. Exclusion criteria included individuals who could not adequately communicate with the research team due to cognitive impairment; behavioral problems including psychosis, neurological damage, or acute intoxication; and deafness, mutism, or aphasia.

Between 2016 and 2018, a team of research assistants visited the homes of all 1,795 eligible adults to request study participation and obtain written informed consent. After providing consent, participants were interviewed in a private location, often in or near the participant's home. Interviews were conducted in Runyankore, the local language. Data were collected using the Computer-Assisted Survey Information Collection (CASIC) Builder software program. Questionnaires had built-in logic and skip patterns based on participant responses. All instruments were written in English, translated into Runyankore, and back translated into English in an iterative process to confirm translation fidelity.

## Measures

ACEs were assessed using a modified version of the Adverse Childhood Experiences–International Questionnaire (ACE-IQ; S1 Text) [54]. The original ACEs instrument was established in the United States (US) [7]. The ACE-IQ was later developed to increase cultural applicability beyond the US context and capture experiences unique to various international settings [55,56]. To date, the ACE-IQ has been validated for use with Malawian adolescents [57] and for adolescents and adults in Nigeria [58].

We modified the ACE-IQ following an iterative process of focus group discussion with key informants. Some of the original ACE-IQ items were dropped because they were either poorly understood or thought to be less applicable for our population. Additional items on food and water insecurity were added because both food and water insecurity are common in this rural setting [40–42]. The modified ACE-IQ included 16 items about exposures to adverse experiences during the participant's first 18 years of life. For some items, participants were probed for frequency of the experience. For the purposes of this analysis, however, all items on the instrument were converted into binary variables, with any experience of the event categorized as 1 and no experience of the event categorized as 0.

The 16 items on the ACE-IQ were grouped into 9 types of ACEs: (1) physical abuse; (2) verbal or emotional abuse; (3) attempted or enacted sexual abuse; (4) residence with an adult who used alcohol or drugs; (5) residence with an adult who had mental illness or who attempted suicide; (6) parents separated or divorced; (7) residence with an adult who was sent to prison or jail; (8) observed violence toward mother or grandmother; and (9) food and/or water insecurity. These 9 binary variables were summed to calculate the cumulative number of ACEs (S2 Text).

Depression symptom severity was measured using the Hopkins Symptom Checklist for Depression (HSCL-D). The self-report instrument assesses for symptoms of depression over the past week and has been modified and validated for use among Runyankore-speaking populations [59,60]. The original HSCL-D includes 15 items. Local modifications involved dropping 1 item ("feeling trapped") and adding 1 item ("don't care what happens to your health") [42,60]. For each item, respondents are asked the frequency of the respective symptom (i.e., not at all, not much, much, very much). The total score on the HSCL-D is calculated as the average of the responses. One question asks the participant to describe how often, during the past 7 days, they had thoughts of ending their life. This item was converted into a binary variable. Participants who responded that they thought about ending their life "much" or "very much" were categorized as having significant suicidal ideation [61].

Because depression screening instruments can often yield overestimates of depression prevalence [62,63], we applied a previously developed algorithm to identify major depressive disorder based on the *Diagnostic and Statistical Manual of Mental Disorders* [64–66]. The 15 items of the HSCL-D were then categorized into the 9 DSM "A" criteria: (A1) depressed mood; (A2) diminished interest or pleasure; (A3) significant weight loss or change in appetite; (A4) insomnia or hypersomnia; (A5) psychomotor agitation; (A6) fatigue or loss of energy; (A7) feeling worthless or guilty; (A8) diminished ability to think or concentrate; and (A9) recurrent thoughts of death. Participants who experienced an HSCL-D item "much" or "very much" in the past week were classified as meeting the respective DSM "A" criteria. We then summed the total number of criteria met, out of a maximum of 9. If a participant reported at least 5 criteria, and met either the A1 criterion (depressed mood) or the A2 criterion (diminished interest or pleasure), s/he was identified as likely meeting diagnostic criteria for a major depressive episode or major depressive disorder.

The demographics questionnaire included questions on age, sex, highest level of educational attainment, marital status, self-reported HIV status (if known), and household asset wealth [67]. Participants were classified into wealth quintile categories (i.e., poorest to richest) based on overall household assets [68,69].

## Ethics

This study received ethical approval from the Mbarara University of Science and Technology Research and Ethics Committee and the Partners Human Research Committee. Consistent with national guidelines, we obtained clearance to conduct the study from the Uganda National Council for Science and Technology.

## Data analysis

The analysis was not preregistered, but we followed a prespecified analysis plan and tracked any deviations that resulted from peer review (S3 Text). To estimate the bivariate associations between the cumulative number of ACEs and depression symptom severity based on the HSCL-D, we fitted a linear regression model to the data with the cumulative number of ACEs as the sole explanatory variable. We then refitted the model, adjusting for the following covariates: sex, age, marital status, primary education completion, HIV status, and household asset wealth quintile category. To determine the extent to which the associations remained statistically significant across the age range, we first fitted a multivariable linear regression model containing a product term between the cumulative number of ACEs and age, specified as a continuous variable. We then conducted further analyses stratified by age, with participants categorized into 1 of 3 age bins: younger adults (26 years of age and younger), adults (27 to 39 years of age), and older adults (40 years of age and older).

We conducted similar analyses to estimate the unadjusted and adjusted associations between the cumulative number of ACEs and major depressive disorder and suicidal ideation. For these analyses, we fitted Poisson regression models with cluster-correlated robust estimates of variance. When applied to binary dependent variable data, the modified Poisson regression model has been shown to yield estimated incidence rate ratios that can be interpreted straightforwardly as relative risk ratios [70].

In secondary analyses, we grouped the cumulative ACEs score into 4 categories for comparability with prior work [26]: no ACEs or 1 ACE (lowest), 2 or 3 ACEs (low), 4 or 5 ACEs (high), and 6 or more ACEs (highest). These ACEs categories were determined based on the interquartile range (IQR) of participants' cumulative ACEs scores. Using this specification of the ACEs categories, we fitted the same linear and Poisson regression models as described above. Next, we disaggregated the cumulative ACEs score into each of the 9 types of ACEs, allowing us to estimate the associations between each type of ACE and depression symptom severity, major depressive disorder, and suicidal ideation, without assuming that the different ACEs had equivalent associations.

To probe the robustness of our findings to confounding by unobserved variables, we used methods proposed by Vanderweele and Ding [71]. We calculated the e-value to determine the minimum strength of association on the risk ratio scale that would be required for an unobserved confounder to have with both the exposure (ACEs) and outcome (depression or suicidality), conditional on the measured covariates, in order to explain away the observed associations. Thus, the e-value quantifies the extent to which unobserved confounding might contribute to the findings.

Following general econometric guidance [72,73], all regression models included adjustment for clustering at the village level. In the setting of multiple levels of clustering, consistent confidence intervals (CIs) will be obtained by using cluster-robust variance estimated at the highest level of clustering. In a sensitivity analysis, we refitted the primary regression models adjusting for clustering at the household level. All analyses were conducted using Stata version 16 (College Station, Texas).

## Results

### Participants

Of 1,795 eligible adult community members (91% response rate), 1,626 were included in the analysis. Slightly over half of the population were women (908 [56%]), and the median age among the 1,602 participants who reported their age was 37 years (IQR 26 to 50), including one 17-year-old emancipated minor. A majority (975 [60%]) of the population had completed at least a primary education. Most adults were either married or cohabiting (993 [61%]), with smaller subsets either separated, divorced, or widowed (288 [18%]) or single/never married (344 [21%]) (Table 1).

Overall, 1,458 participants (90%) had experienced at least 1 ACE before age 18. The median number of ACEs was 3 (IQR, 2 to 5). On average, men reported more ACEs compared with women (3.3 versus 3.28; t = 1.42, $p$ = 0.16), although the difference was not statistically significant. The majority of participants reported physical abuse, verbal or emotional abuse, and residence with an adult who used alcohol or drugs. While the prevalence of most ACEs was comparable across sexes, women were more likely to report experiences of attempted or enacted sexual abuse compared with men, while men were more likely to report verbal or emotional abuse (Table 2).

The mean score on the HSCL-D was 1.48 (standard deviation [SD], 0.42). Women had higher depression symptom severity scores compared with men (1.57 versus 1.38; t = −9.36, $p$ < 0.001). Using a score above 1.75 as a cutoff, 331 participants (20%) screened positive for probable

**Table 1. Sociodemographic and health characteristics of the sample, by sex.**

| | Sex | | | | | |
|---|---|---|---|---|---|---|
| | Female (*n* = 908, 55.8%) | | Male (*n* = 718, 44.2%) | | Total (*n* = 1,626) | |
| | *n* | | *n* | | *n* | |
| **Age:** | | | | | | |
| Young adults (17–26 years) | 224 | 24.7% | 185 | 25.8% | 409 | 25.2% |
| Adults (27–39 years) | 274 | 30.2% | 214 | 29.8% | 488 | 30.0% |
| Older adults (40+ years) | 391 | 43.2% | 314 | 43.7% | 705 | 43.4% |
| Missing | 19 | 2.09% | 5 | 0.70% | 24 | 1.48% |
| **Education:** | | | | | | |
| Completed Primary School | 486 | 53.5% | 489 | 68.1% | 975 | 60.0% |
| Did Not Complete Primary School | 422 | 46.5% | 229 | 31.9% | 651 | 40.0% |
| **Married:** | | | | | | |
| Yes | 526 | 57.9% | 467 | 65.0% | 933 | 61.1% |
| No | 382 | 42.1% | 251 | 35.0% | 633 | 38.9% |
| **Religion:** | | | | | | |
| Protestant | 628 | 69.2% | 502 | 69.9% | 1,130 | 69.5% |
| Catholic | 211 | 23.2% | 174 | 24.2% | 385 | 23.7% |
| Born Again Pentecostal | 57 | 6.38% | 29 | 4.04% | 86 | 5.29% |
| Muslim | 10 | 1.10% | 9 | 1.25% | 19 | 1.17% |
| Other | 2 | 0.002% | 4 | 0.01% | 6 | 0.004% |
| **HIV Status:** | | | | | | |
| HIV–Positive | 108 | 11.9% | 59 | 8.22% | 167 | 10.3% |
| HIV–Negative | 800 | 88.1% | 659 | 91.8% | 1,459 | 89.7% |
| **ACE Category:** | | | | | | |
| Lowest (0–1 ACE) | 218 | 24.0% | 169 | 23.5% | 387 | 23.8% |
| Low (2–3 ACEs) | 301 | 33.2% | 199 | 27.7% | 500 | 30.8% |
| High (4–5 ACEs) | 237 | 26.1% | 212 | 29.5% | 449 | 27.6% |
| Highest (6+ ACEs) | 152 | 16.7% | 138 | 19.2% | 290 | 17.8% |
| **Probable Depression** (HSCL-D > 1.75): | | | | | | |
| Yes | 239 | 26.3% | 92 | 12.8% | 331 | 20.4% |
| No | 669 | 73.7% | 626 | 87.2% | 1,295 | 79.6% |
| **Major Depressive Disorder**: | | | | | | |
| Yes | 114 | 12.6% | 45 | 6.27% | 159 | 9.78% |
| No | 794 | 87.4% | 673 | 93.7% | 1,467 | 90.2% |
| **Suicidal Ideation:** | | | | | | |
| Yes | 22 | 2.42% | 6 | 0.84% | 28 | 1.72% |
| No | 886 | 97.6% | 712 | 99.2% | 1,598 | 98.3% |

ACEs, adverse childhood experiences; HSCL-D, Hopkins Symptom Checklist for Depression.

[a]Figures do not add to 100% due to rounding.

depression. However, only 159 participants (10%) met criteria for probable major depressive disorder. Twenty-eight participants (1.7%) provided responses indicative of suicidal ideation.

## Analyses

In the linear regression model estimating the association between cumulative ACEs score and depression symptom severity, the cumulative number of ACEs was associated with depression

**Table 2. Prevalence of types of ACEs by sex.**

| | Sex | | | | | |
|---|---|---|---|---|---|---|
| | Female (*n* = 908, 55.8%) | | Male (*n* = 718, 44.2%) | | Total (*n* = 1,626) | |
| | *n* | | *n* | | *n* | |
| **Physical Abuse:** | | | | | | |
| Yes | 462 | 50.9% | 362 | 50.4% | 824 | 50.7% |
| No | 446 | 49.1% | 356 | 49.6% | 802 | 49.3% |
| **Verbal or Emotional Abuse:** | | | | | | |
| Yes | 531 | 58.5% | 453 | 63.1% | 984 | 60.5% |
| No | 377 | 41.5% | 265 | 36.9% | 642 | 39.5% |
| **Attempted or Enacted Sexual Abuse:** | | | | | | |
| Yes | 150 | 16.5% | 75 | 10.5% | 225 | 13.8% |
| No | 758 | 83.5% | 643 | 89.6% | 1,401 | 86.2% |
| **Residence with an Adult Who Used Alcohol or Drugs:** | | | | | | |
| Yes | 508 | 56.0% | 420 | 58.5% | 928 | 57.1% |
| No | 400 | 44.1% | 298 | 41.5% | 698 | 42.9% |
| **Residence with an Adult Who Had Mental Illness or Who Attempted Suicide:** | | | | | | |
| Yes | 227 | 25.0% | 181 | 25.2% | 408 | 25.1% |
| No | 681 | 75.0% | 537 | 74.8% | 1,218 | 74.9% |
| **Parents Separated or Divorced:** | | | | | | |
| Yes | 224 | 24.7% | 183 | 25.5% | 407 | 25.0% |
| No | 684 | 75.3% | 535 | 74.5% | 1,219 | 75.0% |
| **Residence with an Adult Who was Sent to Prison or Jail:** | | | | | | |
| Yes | 339 | 37.3% | 274 | 38.2% | 613 | 37.7% |
| No | 569 | 62.7% | 444 | 61.8% | 1,013 | 62.3% |
| **Observed Violence Toward Mother or Grandmother:** | | | | | | |
| Yes | 284 | 31.3% | 234 | 32.6% | 518 | 31.9% |
| No | 624 | 68.7% | 484 | 67.4% | 1,108 | 68.1% |
| **Food and/or Water Insecurity:** | | | | | | |
| Yes | 252 | 27.8% | 284 | 39.6% | 536 | 33.0% |
| No | 656 | 72.3% | 434 | 60.5% | 1,090 | 67.0% |

ACEs, adverse childhood experiences.

[a]Figures do not add to 100% due to rounding

symptom severity (b = 0.046; 95% CI, 0.038 to 0.053; $p < 0.001$) (S1 Table). After adjusting for covariates, the estimated association remained statistically significant (b = 0.050; 95% CI, 0.039 to 0.061; $p < 0.001$) (Table 3). Female sex, older age, and being unmarried were also associated with depression symptom severity in the multivariable regression model. Each additional ACE was associated with 0.050/0.42 = 0.119 SD units of increase in depression symptom severity. A 1 SD difference in the cumulative ACEs score was associated with a 2.20 × 0.119 = 0.26 SD difference in depression symptom severity. In sensitivity analyses, the estimates remained qualitatively similar when we clustered the standard errors at the household level (S2 Table).

In a multivariable linear regression model containing a product term between the cumulative number of ACEs and age, specified as a continuous variable, the estimated regression coefficient on the product term was statistically significant and negative, suggesting an interaction (b = −0.001; 95% CI, −0.0014 to −0.0006; $p < 0.001$): The estimated association between ACEs and depression symptom severity weakened with age (S3 Table). Consistent with this regression model, when we fit regression models stratified by age bin, the estimated association

**Table 3. Adjusted linear and Poisson regression models estimating associations between number of ACEs and depression symptom severity, major depressive disorder, and suicidal ideation.**

| | Depression Symptom Severity | | Major Depressive Disorder | | Suicidal Ideation | |
|---|---|---|---|---|---|---|
| | Adjusted b (95% CI) | p-value | Adjusted RR (95% CI) | p-value | Adjusted RR (95% CI) | p-value |
| **Cumulative No. ACEs** | 0.050 (0.039–0.061) | <0.001 | 1.190 (1.109–1.276) | <0.001 | 1.146 (1.001–1.311) | 0.048 |
| **Female** | 0.178 (0.135–0.222) | <0.001 | 1.858 (1.298–2.660) | 0.001 | 2.620 (1.510–4.544) | 0.001 |
| **Age (years)** | 0.003 (0.002–0.004) | 0.001 | 1.002 (0.995–1.010) | 0.511 | 0.989 (0.959–1.019) | 0.466 |
| **Completed Primary School** | −0.087 (−0.174–0.001) | 0.051 | 0.471 (0.293–0.760) | 0.002 | 0.365 (0.210–0.636) | <0.001 |
| **Married** | −0.068 (−0.113−−0.023) | 0.009 | 0.712 (0.528–0.959) | 0.026 | 1.178 (0.534–2.600) | 0.684 |
| **HIV–Positive** | −0.035 (−0.100–0.030) | 0.239 | 0.809 (0.604–1.085) | 0.158 | 1.071 (0.527–2.176) | 0.849 |
| **Wealth Quintile Category** | | | | | | |
| Poorest | | | | | | |
| Second | −0.060 (−0.123–0.004) | 0.061 | 0.783 (0.547–1.120) | 0.181 | 0.311 (0.077–1.254) | 0.101 |
| Third | −0.040 (−0.102–0.021) | 0.165 | 0.804 (0.609–1.060) | 0.122 | 0.856 (0.379–1.932) | 0.709 |
| Fourth | −0.068 (−0.123−−0.013) | 0.022 | 0.644 (0.461–0.902) | 0.010 | 0.284 (0.066–1.214) | 0.089 |
| Richest | −0.035 (−0.124–0.055) | 0.390 | 1.008 (0.651–1.562) | 0.970 | 1.019 (0.414–2.508) | 0.967 |
| **Constant** | 1.246 (1.177–1.315) | <0.001 | 0.065 (0.030–0.139) | <0.001 | 0.018 (0.004–0.081) | <0.001 |
| **Observations** | 1,602 | | 1,602 | | 1,602 | |
| **R² and Pseudo R²** | 0.157 | | 0.076 | | 0.080 | |

ACEs, adverse childhood experiences; b, beta coefficient; CI, confidence interval; RR, relative risk.

Each model is adjusted for sex, age, primary school completion, marital status, HIV status, and household asset wealth quintile category.

between the cumulative number of ACEs and depression symptom severity was largest among younger adults (b = 0.060; 95% CI, 0.047 to 0.073; $p < 0.001$) and adults (b = 0.064; 95% CI, 0.056 to 0.071; $p < 0.001$) and weakest among older adults (b = 0.031; 95% CI, 0.017 to 0.046; $p = 0.001$).

In the multivariable Poisson regression models, an increase in the cumulative number of ACEs was associated with major depressive disorder (adjusted relative risk [ARR] = 1.190; 95% CI, 1.109 to 1.276; $p < 0.001$). Similarly, the cumulative number of ACEs was associated with suicidal ideation (ARR = 1.146; 95% CI, 1.001 to 1.311; $p = 0.048$).

The linear regression models estimating the associations between categorical ACEs score and depression symptom severity indicated a graded increase in associations between ACEs category and depression symptom severity, with the fourth category (6 or more ACEs) associated with the highest depression symptom severity score, both before (b = 0.287; 95% CI, 0.227 to 0.346; $p < 0.001$) (S4 Table) and after adjusting for covariates (b = 0.313; 95% CI, 0.242 to 0.384; $p < 0.001$) (Table 4). Study participants who reported 6 or more ACEs had 0.313/ 0.42 = 0.75 greater SD units of depression symptom severity compared with study participants who reported 0 to 1 ACE. The Poisson regression models estimating the association between categorical ACEs score and major depressive disorder also indicated a graded increase in the prevalence of depression across ACE categories. Individuals who had experienced 6 or more ACEs during childhood were over 2 and a half times as likely to meet criteria for major depressive disorder (RR = 2.616; 95% CI, 1.725 to 3.967; $p < 0.001$; ARR = 2.819; 95% CI, 2.030 to 3.916; $p < 0.001$) as those who reported 0 to 1 ACE. None of the categorical ACEs scores, however, had a statistically significant association with suicidal ideation (e.g., ≥6 ACEs: RR = 2.669; 95% CI, 0.973 to 7.320; $p = 0.057$; ARR = 2.340; 95% CI, 0.844 to 6.488; $p = 0.102$).

Linear regression models demonstrated statistically significant associations between each of the 9 types of ACE and depression symptom severity (S5 Table). Th-estimated associations remained statistically significant after adjusting for covariates. In the adjusted models,

**Table 4. Adjusted linear and Poisson regression models estimating associations between ACE category and depression symptom severity, major depressive disorder, and suicidal ideation.**

| | Depression Symptom Severity | | Major Depressive Disorder | | Suicidal Ideation | |
|---|---|---|---|---|---|---|
| | Adjusted b (95% CI) | p-value | Adjusted RR (95% CI) | p-value | Adjusted RR (95% CI) | p-value |
| **ACE Category** | | | | | | |
| Lowest (0–1 ACE) | | | | | | |
| Low (2–3 ACEs) | 0.081 (0.008–0.155) | 0.035 | 1.356 (0.961–1.913) | 0.083 | 1.668 (0.664–4.191) | 0.276 |
| High (4–5 ACEs) | 0.138 (0.070–0.205) | 0.002 | 1.595 (1.137–2.236) | 0.007 | 1.141 (0.464–2.802) | 0.774 |
| Highest (≥6 ACEs) | 0.313 (0.242–0.384) | <0.001 | 2.819 (2.030–3.916) | <0.001 | 2.340 (0.844–6.488) | 0.102 |
| **Female** | 0.178 (0.135–0.222) | <0.001 | 1.860 (1.287–2.689) | 0.001 | 2.531 (1.434–4.467) | 0.001 |
| **Age (years)** | 0.003 (0.001–0.004) | 0.001 | 1.003 (0.995–1.010) | 0.492 | 0.989 (0.957–1.021) | 0.491 |
| **Completed Primary School** | −0.088 (−0.178–0.001) | 0.052 | 0.468 (0.287–0.764) | 0.002 | 0.357 (0.200–0.635) | <0.001 |
| **Married** | −0.066 (−0.111–−0.020) | 0.011 | 0.715 (0.533–0.960) | 0.026 | 1.199 (0.565–2.541) | 0.637 |
| **HIV–Positive** | −0.034 (−0.101–0.032) | 0.260 | 0.802 (0.606–1.062) | 0.124 | 1.076 (0.544–2.128) | 0.833 |
| **Wealth Quintile Category** | | | | | | |
| Poorest | | | | | | |
| Second | −0.062 (−0.125–0.002) | 0.055 | 0.783 (0.546–1.122) | 0.182 | 0.321 (0.079–1.303) | 0.112 |
| Third | −0.041 (−0.106–0.023) | 0.171 | 0.792 (0.597–1.051) | 0.106 | 0.891 (0.398–1.993) | 0.778 |
| Fourth | −0.072 (−0.128–−0.016) | 0.019 | 0.624 (0.455–0.855) | 0.003 | 0.283 (0.066–1.218) | 0.090 |
| Richest | −0.038 (−0.124–0.049) | 0.339 | 0.984 (0.645–1.502) | 0.941 | 1.014 (0.417–2.468) | 0.975 |
| **Constant** | 1.300 (1.236–1.365) | <0.001 | 0.079 (0.035–0.176) | <0.001 | 0.020 (0.004–0.112) | <0.001 |
| **Observations** | 1,602 | | 1,602 | | 1,602 | |
| **$R^2$ and Pseudo $R^2$** | 0.149 | | 0.071 | | 0.081 | |

ACE, adverse childhood experience; b, beta coefficient; CI, confidence interval; RR, relative risk.

Each model is adjusted for sex, age, education, marital status, HIV status, and household asset wealth quintile category.

associations were strongest for attempted or enacted sexual abuse (b = 0.193; 95% CI, 0.127 to 0.259; $p < 0.001$), observing violence toward the mother or grandmother (b = 0.131; 95% CI, 0.088 to 0.174; $p < 0.001$), and food and/or water insecurity (b = 0.187; 95% CI, 0.137 to 0.238; $p < 0.001$). In the adjusted models, every ACE had a statistically significant association with major depressive disorder. In the models estimating associations between each ACE and suicidal ideation, however, the only 2 experiences that had statistically significant associations with suicidal ideation were residence with an adult who was sent to prison or jail (ARR = 2.654; 95% CI, 1.646 to 4.278; $p < 0.001$) and food and/or water insecurity (ARR = 1.882; 95% CI, 1.155 to 3.065; $p = 0.011$).

We explored the robustness of our findings to potential confounding from unobserved variables. Using as an example the multivariable Poisson regression estimate for the association between the highest category of ACEs exposure and major depressive disorder, we obtained an e-value of 5.08. Thus, an unobserved confounder would need to have a strength of association, on the risk ratio scale, with the highest category of ACEs exposure and with major depressive disorder of 5.08 each to move our estimated association to include a risk ratio of 1.

## Discussion

In this cross-sectional, population-based study of adults in rural Uganda, we demonstrated robust associations between cumulative number of ACEs and depression symptom severity, major depressive disorder, and suicidal ideation. Furthermore, we estimated a graded association, whereby the prevalence of depression was highest among individuals who reported the

highest category of ACEs (≥6 experiences). Significant associations were present for all 9 types of ACEs and depression, and between 2 of the ACEs and suicidal ideation.

While previous research from sub-Saharan Africa has shown associations between ACEs and depression among adolescents and young adults [30–37], our findings demonstrate that the associations are consistent across all age ranges within a general population of adults. Moreover, our study extends prior work by assessing major depressive disorder and suicidal ideation in addition to using a standardized depression screening instrument. The estimated associations between ACEs and major depressive disorder and suicidal ideation are reflective of research showing strong associations between high levels of adversity and major depressive disorder among older adults [74], as well as research indicating associations between ACEs and suicidal ideation and suicide attempts across the lifespan [13,75]. Additionally, the finding that participants who reported food and water insecurity during childhood were at higher risk for suicidal ideation is consistent with a previously published multicountry study showing an increased odds of suicide attempt among adolescents reporting severe food insecurity [76]. Finally, the results suggest that our findings are unlikely to be completely explained by confounding from unobserved variables.

## Limitations

These findings must be considered in the context of existing limitations. The reliance on self-report of ACEs during adulthood presents challenges, as retrospective self-report can be subject to recall bias. As Baldwin and colleagues (2019) identified in their systematic review and meta-analysis, retrospective and prospective accounts of childhood adversity can be inconsistent [77]. A study from the South African Birth to Twenty cohort found low concordance between adolescent and young adults' retrospective reports of ACEs and their caregivers' prospective reports [78]. If people with depression are more likely to recall ACEs during childhood, then this measurement error could bias our estimated associations away from the null. A second limitation is that, due to the cross-sectional nature of this study, we are unable to infer causal relations between these interrelated factors. However, our estimates were robust to different specifications, and the e-value analysis suggests that an unmeasured confounder would need to be very strongly associated with both ACEs and depression in order to fully explain away the observed associations.

A third limitation, common to nearly all studies using the ACEs questionnaire, is the lack of detail regarding each experience [79]. For example, one question elicits whether a family member or household-dwelling adult went to prison or jail while the participant was a child or adolescent. However, the instrument does not probe for details about the relationship between the study participant and this adult figure, including the quality of their relationship or the importance of this adult figure's role in their life. Another consideration is the effect of age on reactions to stressful situations. The current instrument does not ask participants how old they were when the experience occurred.

Fourth, our survey instrument may have yielded underestimates of some of the ACEs. Some of the participants lived with their parents or grandparents in intergenerational households, and such living arrangements could have limited our ability to accurately collect sensitive data. However, research assistants ensured that interviews were conducted in a private location out of earshot of other members of the household. Somewhat related to this limitation, physical discipline is normative in East Africa [80,81]. Responses to the physical abuse questions may potentially underrepresent the extent of the experience if participants responded negatively to those ACE-IQ questions, i.e., because they perhaps believed such behaviors to be standard practice. Although these limitations could have caused us to

underestimate the prevalence of certain ACEs, they would only have biased our estimates of the associations between ACEs and the mental health outcomes if the factors leading participants to underreport ACEs were also associated with the mental health outcomes.

Lastly, while this study employed a modified ACE-IQ adapted for the local context, the questionnaire focused primarily on experiences encountered within the household and/or perpetrated by a household-dwelling adult. However, research has demonstrated pervasive violence toward children in Uganda enacted outside the household and perpetrated by other individuals, including school staff, peers, neighbors, and strangers [43,82].

## Implications for research, clinical Practice, and public policy

To address limitations in our research, future work may use prospective study designs such as that exemplified by Cluver and colleagues in South Africa [33]. Additionally, studies may involve mixed methods, incorporating qualitative interviews that probe for additional detail on ACEs. By developing a better understanding of how developmental stage interacts with ACEs to influence later outcomes, we can better adapt intervention programs for children and their caregivers. Finally, future studies should assess abuse and other traumatic events experienced both within and outside the household. These findings can thereby be used to inform policies and practice to address violence perpetrated at multiple levels within the community.

To address individual-level vulnerability and reduce the "stressogenic" environments that put individuals at higher risk of childhood adversity and adult depression [25], programs and policies are needed to provide support for children, adolescents, and their families. Preventive programs in schools may focus on supporting natural protective factors among children. Furthermore, socioeconomic interventions may provide additional support to parents during stressful economic times. A systematic review found that 35% of socioeconomic interventions, including housing, conditional cash transfer, and income supplementation, reduced children and adolescents' exposure to ACEs [83]. Research from LMICs has similarly demonstrated positive effects of cash handouts on health, nutrition, school attendance, and cognitive development [84]. While individual-, family-, and neighborhood-level programs are important to addressing abuse, household dysfunction, and neglect, the high level of childhood adversity seen among this population needs to be recognized as a wider public health policy issue.

While it is vital that future policies and programs address high rates of ACEs in this region, many ACEs reflect larger structural and systematic barriers in Uganda, including poverty and economic insecurity. Thus, while targeting specific ACEs to prevent child and adult psychopathology [85], there remains a need to address depression, suicidal ideation, and other associated negative mental health outcomes. Due to the cross-sectional nature of this study, we were unable to determine causality between ACEs and poor mental health during adulthood. For example, adults may react differently to children who have mental health problems during childhood and/or adolescence, including both internalizing and externalizing disorders, thereby predisposing these children and adolescents to certain types of ACEs. Furthermore, early mental health problems may modify individuals' perceptions of the world and their experiences. As we are not able to rule out reverse causality as a potential explanation for the observed associations, the present findings may have implications for mental health care among both children and adults. Considering high rates of childhood adversity, the high prevalence of depression, and pervasive barriers to accessing behavioral and mental health care across the country, it is critical that mental health be given more recognition and attention within the public health and health systems agenda in Uganda [47,49,86]. Improvements in behavioral and mental health care may in turn provide support for child caregivers and thereby reduce future ACEs [87].

## Conclusions

Given the potential relationship between ACEs and adult depression, interventions are needed to prevent childhood adversity and respond to health and social systems in this context. By addressing multilevel factors contributing to these experiences, programs may reduce ACEs among children and adolescents. Based on our interpretation of the present findings, such intervention may improve trajectories toward poor mental health during adulthood. Given high rates of depression and challenges associated with reducing certain ACEs in this context, programs must also be developed to address barriers to accessing mental health and psychosocial support services.

## Supporting information

**S1 Checklist. STROBE checklist.**
(DOCX)

**S1 Text. Modified Adverse Childhood Experiences–International Questionnaire (ACE-IQ).**
(DOCX)

**S2 Text. Calculating the cumulative ACEs score from the modified version of the ACE-IQ–Binary Version.**
(DOCX)

**S3 Text. Methods.**
(DOCX)

**S1 Table. Unadjusted linear and Poisson regression models estimating associations between cumulative number of ACEs and depression symptom severity, major depressive disorder, and suicidal ideation.**
(DOCX)

**S2 Table. Adjusted linear and Poisson regression models estimating associations between number of ACEs and depression symptom severity, major depressive disorder, and suicidal ideation, using standard errors clustered at the household level.**
(DOCX)

**S3 Table. Linear regression model with product term between cumulative number of ACEs and age (specified as a continuous variable), and linear regression models estimating associations between cumulative number of ACEs and depression symptom severity, stratified by age category.**
(DOCX)

**S4 Table. Unadjusted linear and Poisson regression models estimating associations between ACEs category and depression symptom severity, major depressive disorder, and suicidal ideation.**
(DOCX)

**S5 Table. Linear and Poisson regression models estimating associations between each type of ACE and depression symptom severity, major depressive disorder, and suicidal ideation.**
(DOCX)

## Acknowledgments

We thank the HopeNet cohort study participants, without whom this research would not be possible. We also thank members of the HopeNet study team for research assistance; in addition to the named study authors, HopeNet and other collaborative team members who contributed to data collection and/or study administration during all or any part of the study were as follows: Phionah Ahereza, Owen Alleluya, Patience Ayebare, Dickson Beinomugisha, Bridget Burns, Patrick Gumisiriza, Clare Kamagara, Justus Kananura, Viola Kyokunda, Juliet Mercy, Patrick Lukwago Muleke, Rhina Mushagara, Rumbidzai Mushavi, Moran Owembabazi, Sarah Nabachwa, Immaculate Ninsiima, Mellon Tayebwa, and Dagmar Vořechovská. We also thank Roger Hofmann of West Portal Software Corporation (San Francisco, Calif.), for developing and customizing the Computer Assisted Survey Information Collection Builder (TM) software program used for survey administration.

The content is solely the responsibility of the authors and does not necessarily represent the views of Friends of a Healthy Uganda or US National Institutes of Health.

## Author Contributions

**Conceptualization:** Bernard Kakuhikire, Alexander C. Tsai.

**Data curation:** Emily N. Satinsky.

**Formal analysis:** Emily N. Satinsky.

**Funding acquisition:** Bernard Kakuhikire, Alexander C. Tsai.

**Investigation:** Emily N. Satinsky, Bernard Kakuhikire, Charles Baguma, Justin D. Rasmussen, Scholastic Ashaba, Christine E. Cooper-Vince, Jessica M. Perkins, Allen Kiconco, Elizabeth B. Namara, David R. Bangsberg, Alexander C. Tsai.

**Project administration:** Emily N. Satinsky, Bernard Kakuhikire, Charles Baguma, Justin D. Rasmussen, Alexander C. Tsai.

**Supervision:** Alexander C. Tsai.

**Writing – original draft:** Emily N. Satinsky.

**Writing – review & editing:** Bernard Kakuhikire, Charles Baguma, Justin D. Rasmussen, Scholastic Ashaba, Christine E. Cooper-Vince, Jessica M. Perkins, Allen Kiconco, Elizabeth B. Namara, David R. Bangsberg, Alexander C. Tsai.

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
