## [Editor Report · Decision Letter 0]

17 Jun 2020

Dear Dr Satinsky, 

Thank you for submitting your manuscript entitled "Associations between adverse childhood experiences and adult depression symptom severity, major depressive disorder, and suicidal ideation in rural Uganda: A cross-sectional, population-based study" for consideration by PLOS Medicine.

Your manuscript has now been evaluated by the PLOS Medicine editorial staff [as well as by an academic editor with relevant expertise] and I am writing to let you know that we would like to send your submission out for external peer review.

Kind regards,

Caitlin Moyer, Ph.D.,

Associate Editor

PLOS Medicine

---

## [Decision Letter · Decision Letter 1]

20 Nov 2020

Dear Dr. Satinsky,

Thank you very much for submitting your manuscript "Associations between adverse childhood experiences and adult depression symptom severity, major depressive disorder, and suicidal ideation in rural Uganda: A cross-sectional, population-based study" (PMEDICINE-D-20-02698R1) for consideration at PLOS Medicine. 

I apologize for the delay in returning a decision on your manuscript. Your paper was evaluated and discussed among all the editors here. It was also sent to three independent reviewers, including a statistical reviewer. The reviews are appended at the bottom of this email and any accompanying reviewer attachments can be seen via the link below:

[LINK]

In light of these reviews, I am afraid that we will not be able to accept the manuscript for publication in the journal in its current form, but we would like to consider a revised version that addresses the reviewers' and editors' comments. Obviously we cannot make any decision about publication until we have seen the revised manuscript and your response, and we plan to seek re-review by one or more of the reviewers. 

We expect to receive your revised manuscript by Dec 11 2020 11:59PM. Please email us (plosmedicine@plos.org) if you have any questions or concerns.

We look forward to receiving your revised manuscript. 

Sincerely,

Caitlin Moyer, Ph.D.

Associate Editor 

PLOS Medicine

plosmedicine.org

1.Data availability statement: Thank you for your willingness to make your data and code available. At this time, please provide information on where the anonymized data may be accessed.

2. Protocol: Did your study have a prospective protocol or analysis plan? Please state this (either way) early in the Methods section.

3. Abstract: Methods and Findings: Early on in this section, please mention the key features of the study design (e.g. population and setting, number of participants, years during which the study took place, and main outcome measures). Please provide some summary demographic information pertaining to the study participants, including how outcome measures were obtained (ACEs numbers, depression symptom severity/MDD diagnosis, suicide ideation).

4. Abstract: Methods and Findings: For the reported results on number of ACEs and depressive symptoms, risk of major depressive disorder, and suicidal ideation, please provide the p values in addition to providing confidence intervals (please also define the abbreviation CI at first use).

5. Abstract: Methods and Findings: Please mention the important variables that are adjusted for in the analyses.

6. Abstract: Methods and Findings: In the last sentence of the Abstract Methods and Findings section, please describe the main limitation(s) of the study's methodology.

7. Abstract: Conclusion: In the first sentence, we suggest you address the results of the study prior to mentioning the implications; the phrase "In this study, we observed ..." may be useful.

8. Author Summary: At this stage, we ask that you include a short, non-technical Author Summary of your research to make findings accessible to a wide audience that includes both scientists and non-scientists. The Author Summary should immediately follow the Abstract in your revised manuscript. This text is subject to editorial change and should be distinct from the scientific abstract. Please see our author guidelines for more information: https://journals.plos.org/plosmedicine/s/revising-your-manuscript#loc-author-summary

9. Throughout the text: Please use square brackets for in-text citations, like this [1].

10. Methods: Line 85: Please remove the trademark symbol.

11: Methods: Line 90: Please provide the modified version of the ACE-IQ and reference it along with your description as a supporting information file.

12. Methods: Line 119-123: As mentioned by a reviewer, please do provide some further description of the algorithm used to identify major depressive disorder, even though the references are provided.

13: Results: Line 199-202: Please provide the p values in addition to the 95% CIs for the unadjusted and adjusted associations between ACEs and depression severity scores.

14: Results: Line 219-221: Please also provide p values in addition to 95% CIs for unadjusted and adjusted associations between major depressive disorder and ACEs number.

15. Results: Line 221-224: Please provide p values in addition to 95% CIs for unadjusted and adjusted associations between suicidal ideation and cumulative ACE score.

16. Results Lines 225-239: for the linear regression models with categorical ACE score, the comparisons between major depressive disorder between ACE categories, and the association between ACE category and suicidal ideation, please provide p values for adjusted and unadjusted results. Please present the unadjusted results (in a table) and reference the location in the text here.

17: Results: Lines 244-255: Please include p values together with 95% CIs for the associations between each of the types of ACE and depression symptoms, and please include a table of the unadjusted analyses.

18. Results: Lines 252-255: Please present the results with 95% CI and p value for the association between having a parent/adult in prison and suicidal ideation, as this is called out in the text.

19. Discussion: Please slightly reorganize the Discussion as follows: a short, clear summary of the article's findings; what the study adds to existing research and where and why the results may differ from previous research; strengths and limitations of the study; implications and next steps for research, clinical practice, and/or public policy; one-paragraph conclusion.

20. Discussion: In the conclusion paragraph, please temper the statements to avoid inferring the direction of the relationship (ACE leads to depression) and policy/health systems recommendations based on this- while this is an interpretation of the findings, given the limitations mentioned and the cross sectional nature of the study, causal implications should be avoided (lines 381-383).

21. Table 1 and Table 2: Please remove the % sign from the columns headed “%” and please define all abbreviations such as ACE and HSCLD, in the legends.

22. Table 3 and Table 4: Please provide the actual p values associated with the adjusted results (rather than noting p<0.05, etc). In the legend, please provide the abbreviations for ACE, and HSCLD-15, and note which variables were adjusted for in these analyses. Please provide the unadjusted results (and reference in the text around Line 201, 220 and 223)- if desired this can be presented as a supporting information file. Please also clarify whether the suicidal ideation column is a comparison with total number of ACE, or cumulative ACE score (as mentioned in the text).

23. S1 Appendix and S2 Appendix: Please note the actual p values, rather than * for p<0.05, for example. Please note in the legend of S1 Appendix what variables were adjusted for. For both, please also provide the unadjusted results.

24. Checklist: Please ensure that the study is reported according to the STROBE guideline, and include the completed STROBE checklist as Supporting Information. When completing the checklist, please use section and paragraph numbers, rather than page numbers. Please add the following statement, or similar, to the Methods: "This study is reported as per the Strengthening the Reporting of Observational Studies in Epidemiology (STROBE) guideline (S1 Checklist)."

Comments from the reviewers:

Reviewer #1: See attachment

Michael Dewey

Reviewer #2: This is a very good paper, thorough and well constructed. It follows closely the established ACE methodology conducted by the CDC in the US, and replicated in other high-income settings. Its findings are perhaps not totally surprising, but they are nonetheless very important. We have always assumed that ACE's work in similar ways in LMIC, but this is a very robust investigation of what is actually an empirical question, and the findings are very clear. In fact, they are so close to the findings of the original CDC study (which had a very similar method - retrospective reporting of a population cohort) that this is worth remarking on. ACE impact people in Sub-Saharan Africa just like they do in America. This does matter and should be published.

I would have been interested to see an analysis by gender, but I don't think this should be required at all - just a thought (and gender is in the models). I should add that I'm not so familiar with the adult mental health literature in Eastern Africa, so I have taken as accurate that this kind of study has not been conducted before in the region. 

I would recommend this paper for publication, and commend the authors. I'd be interested to know what the objections of previous reviewers were. I would also be very interested for future papers to see the impacts of ACEs on other health, economic and social outcomes. Lucie Cluver.

Reviewer #3: Overall, the methodology and writing is solid. To make an impact, it needs to go further: it needs to really contextualize this piece of research in the larger field, it needs to make a case for what new information it is bringing to the field, and it needs to dive deeper into a rich discussion of what those findings mean for practice. As written, the discussion is not sufficiently developed. 

* The literature review seems to neglect a lot of work that has been done on ACEs and psychological health in Africa. For example, Cluver et al in South Africa; Manyema et al with young adults in South Africa, and Kidman et al in Malawi have all explicitly studied ACEs and suicidality, depression or a similar outcomes. These are not perfect parallels to the research question or age group here, but would provide additional context. 

* The introduction states that this line of research has been done in HIC, but not LMIC. However, there is no nuanced discussion of how these contexts might matter to the relationship under study. 

* There is also little on how ACEs get embodied. A sentence or two on how ACEs translate into later depression would be helpful.

* Why was this particular area (Nyakabare Parish) chosen for this study? Why a census instead of a random sample of a larger area? 

* The paper states that the ACE-IQ was modified, but it isn't clear how it was modified. Please provide details. How was it coded? It mentions a point for each item, but typically the coding for the ACE-IQ is a bit more complicated and based at least on a point per type of adversity (which can cover multiple items). Table 2 seems to list these, indicating the range should be 0-9. Is this correct? Later you refer to 16 - but that should be questions not ACEs. And why was food insecurity included? That decision is fine, but a justification is needed. I think some of the other questions are a little different too. This will be confusing when researchers try to compare ACE prevalence across populations/papers without adequate descriptions. 

* It would be helpful to have a bit more on the algorithm for the depression diagnosis in the paper, especially since it is the key outcome. 

* Analyses adjust for clustering at the village level, but what about the household level? If you only interviewed one per household, this should be in the methods. However, based on the numbers presented, I don't think this is the case. 

* Do you have the statistical power to examine individual ACEs and suicide ideation? 

* What value do you get out of the individual ACE models? Given the similarity in coefficients, are they important alone, or do they serve as a proxy for overall ACE exposure? For these analyses to be included, it would be essential to motivate them the introduction. Why isn't a cumulative score a better indicator; why would individual ACEs matter and which ones would you expect to see generate a large association? In the absence of such, this comes off as rather exploratory. 

* The paragraph starting with line 275 seems to fit better in an introduction. I am not clear how it sheds further light on the new findings. I am also not clear where poverty comes into the discussion of ACEs and depression. 

* The last sentence of that paragraph implies that the study found little access to support or resources; this isn't reported in the paper. 

* The paragraph starting with line 289 also seems out of place. This is background material, and better suited for an introduction. It is not again related to the findings. The focus on "stressogenic" environments and poverty in particular seems to be introduced for the first time here, and is background - it isn't something we learn from or sheds light on the findings. 

* The discussion on implications seems like it could have been written before this study was conducted. It doesn't answer new questions about what should be done using your new findings. The authors start to bring in some of this - such as the fact that ACEs were incredibly high compared to other contexts - but don't drill down. They also start to acknowledge the limited mental health resources available in Uganda, but don't really go into how the new information on ACEs relates to this. The authors make the argument that the high rates of depression warrant more investment in mental health, but this isn't the core of the paper/findings - in fact, these rates were given as background. What other approaches could be useful in this context, given the new info? If there are Ugandan authors, perhaps they can help.

[LINK]

---

## [Decision Letter · Decision Letter 2]

16 Apr 2021

Dear Dr. Satinsky,

Thank you very much for re-submitting your manuscript "Associations between adverse childhood experiences and adult depression symptom severity, major depressive disorder, and suicidal ideation in rural Uganda: cross-sectional, population-based study" (PMEDICINE-D-20-02698R2) for review by PLOS Medicine.

I have discussed the paper with my colleagues and the academic editor and it was also seen again by one of the original reviewers. I am pleased to say that provided the remaining editorial and production issues are dealt with we are planning to accept the paper for publication in the journal.

[LINK]

We look forward to receiving the revised manuscript by Apr 23 2021 11:59PM.   

Sincerely,

Caitlin Moyer, Ph.D.

Associate Editor 

PLOS Medicine

plosmedicine.org

Requests from Editors:

1.From the academic editor: Methods: Line 133-134: Is it relevant to say that the study is part of a population based cohort? Please clarify to avoid confusion.

2. From the academic editor: Methods: Line 200-201: Although the Hopkins Symptom Checklist has been validated in Uganda, please confirm whether it has been validated in this Ugandan language or cultural group (or in a rural setting).

3. We suggest revising the title to: “Adverse childhood experiences, adult depression and suicidal ideation in rural Uganda: A cross-sectional, population-based study”

4. Please revise the short title to: “Adverse childhood experiences and adult depression in Uganda”

5. Data availability statement: Please update the statement at this time, with your plan for making the anonymized data and code available ( with links for access to anonymized data and Stata code in a repository or similar).

6. Abstract: Methods and Findings: Line 10: Should “elicited” be “assessed” or similar? (“...ACES were elicited using a modified…”)

7. Abstract: Line 15: Please remove “significant” before suicidal ideation.

8. Introduction: Line 122 (and throughout text): For in-text references, where a range or list of references are noted within brackets, please do not include spaces [31,32].

9. Introduction: Line 122: Please qualify “...no study has yet…” with “to our knowledge” or similar.

10. Methods: Line 159: Please provide the number of adults represented by the 758 households.

11. Methods: Line 168: Please provide the number of eligible adults.

12. Methods: Line 267-268: Please clarify the rationale for not adjusting at the household level, in line with the response to reviewer 3, comment 7. For example, in the response to reviewers it is mentioned that the estimates in Table 3 are consistent between clustering at village vs household level, but you also note that confidence intervals become less precise adjusting for household clustering. Please consider incorporating this into the supporting information as an additional analysis.

13. Results: Line 299: Please remove “significant” before suicidal ideation.

14. Discussion: Thank you for including the sections describing the strengths and limitations of the work, as well as implications for future research, practice, and policy. If possible, please consider expanding on the paragraph between lines 380-386, where you describe your findings in the context of the existing literature.

15. Discussion: Line 377: Please consider changing the wording of “risk of developing depression” as the study is cross-sectional it may be more conservative to avoid implying any temporal nature of the association.

16. Discussion: Line 384-386: Please revise this sentence to make the meaning more clear: “Finally, we demonstrate that our findings are such that only strong confounding by unobserved variables could completely explain them.” This could be modified to “...results suggest that our findings are unlikely to be completely explained by unobserved variables, fo example…” or similar.

17. Discussion: Line 472: Please consider tempering “prevent” to “improve” here, to avoid any causal implications.

18. References: Please use the "Vancouver" style for reference formatting, and see our website for other reference guidelines https://journals.plos.org/plosmedicine/s/submission-guidelines#loc-references

(For example, in Ref 4, PLoS Medicine should be PLoS Med).

19. Supporting Information: Thank you for including the revised supporting information documents. Please submit a finalized version for each item (rather than the tracked changes versions). It would be helpful for each item to be included in an independent file and labeled as such (S1_Analysis Plan, S1_Checklist, S1_Table, etc.)

20. S3 Appendix (Analysis plan): It would be helpful to include in this Prespecified Analysis Plan document a brief mention of the specific outcomes and analyses planned, to go along with your statement that all the described analyses were specified at the outset of the study. Thank you for thoroughly documenting the changes made to the analyses during the course of peer review.

Comments from Reviewers:

Reviewer #1: The authors have addressed my points.

I may not have been sufficiently clear about the quintiles issue. My point was that this is potentially ambiguous as there are only four quintiles. It would have been possible to use them in the regression but that is not what the authors did. I agree this is a bit picky. I suggested quintile categories or even the everyday English word fifths but if the authors insist I would not want to man the barricades over it. I realise I am fighting a losing battle here anyway and will soon have to accept people talking about dividing a ample into two medians.

Michael Dewey

[LINK]

---

## [Editor Report · Decision Letter 3]

29 Apr 2021

Dear Dr Satinsky, 

On behalf of my colleagues and the Academic Editor, Charlotte Hanlon, I am pleased to inform you that we have agreed to publish your manuscript "Adverse childhood experiences, adult depression, and suicidal ideation in rural Uganda: A cross-sectional, population-based study" (PMEDICINE-D-20-02698R3) in PLOS Medicine.

Please also address the following two editorial requests:

- Data availability statement: Thank you for providing the GitHub link. At this time, it seems that only the ReadMe file is posted. Please also upload the data files. Also, please update the Data Availability Statement within the manuscript submission system.

- Reference List: Please double-check the formatting of the References. For example, some information appears to be missing from Reference 14, Reference 34, Reference 35. Please update Reference 78.

PRESS

Sincerely, 

Caitlin Moyer, Ph.D. 

Associate Editor 

PLOS Medicine